# Geometric Morphometric Analysis and Molecular Identification of Coconut Mite, *Aceria guerreronis* Keifer (Acari: Eriophyidae) Collected from Thailand

**DOI:** 10.3390/insects13111022

**Published:** 2022-11-05

**Authors:** Suradet Buttachon, Siwaret Arikit, Wirawan Nuchchanart, Thanapol Puangmalee, Tidapa Duanchay, Nattaya Jampameung, Sunisa Sanguansub

**Affiliations:** 1Department of Entomology, Faculty of Agriculture at Kamphaeng Saen, Kamphaeng Saen Campus, Kasetsart University, Nakhon Pathom 73140, Thailand; 2Department of Agronomy, Faculty of Agriculture at Kamphaeng Saen, Kamphaeng Saen Campus, Kasetsart University, Nakhon Pathom 73140, Thailand; 3Department of Animal Science, Faculty of Agriculture at Kamphaeng Saen, Kasetsart University, Nakhon Pathom 73140, Thailand; 4Center for Agricultural Biotechnology, Kamphaeng Saen Campus, Kasetsart University, Nakhon Pathom 73140, Thailand; 5Center of Excellence on Agricultural Biotechnology, Office of the Permanent Secretary, Ministry of Higher Education, Science, Research and Innovation (AG-BIO/MHESI), Bangkok 10900, Thailand

**Keywords:** acarology, morphometry, molecular techniques, *Cocos nucifera* L.

## Abstract

**Simple Summary:**

The coconut mite is one of the most well-known and serious pests of coconut fruits worldwide, and it has spread to most regions where coconuts are produced; in Thailand, *Aceria guerreronis* Keifer (Acari: Eriophyidae) is a quarantine pest. We conducted a geometric morphometric analysis and molecular identification on coconut mites collected from Thailand to obtain their origin and history. Our findings will provide a genetic resource for future functional studies on the relative phylogenetic relationship of coconut mites, which will be performed to understand how coconut mite species interact with their host plant. These findings will be helpful in designing pest management strategies against quarantine pests in Thailand.

**Abstract:**

One of the most impactful pests in several coconut production regions across the world is the coconut mite, *Aceria guerreronis* Keifer. Scholars can obtain some necessary biogeographic information about coconut mites from studies that explore the geographic patterns of morphological variations and molecular properties among coconut mite populations from various locales. To investigate the geographical origin, ancestral host associations, and colonization history of the mite in Thailand, we obtained DNA sequence data from two mitochondrial (16s and COI) and one nuclear region (ITS) from coconut mite samples originating from 25 populations; additionally, we analyzed the morphological variations in the prodorsal shield and the coxigenital and ventral regions of the mite idiosoma. From the results of experiments using both identification methods, we identified the mite as the coconut mite, *A. guerreronis* (Acari: Eriophyidae). According to the phylogenetic analysis results of the 25 mite samples, we classified the mites as being closely related to mites found by the authors of a previous report in India. We are the first to report the results of a geometric morphometric analysis and molecular identification of *A. guerreronis* in Thailand, and our findings support the idea that the mites’ origin and invasion history are not well documented, which makes it difficult to apply quarantine procedures and search for biological pest control agents.

## 1. Introduction

The invasive coconut mite *Aceria guerreronis* Keifer (Acari: Eriophyidae) has spread and become well established in the main coconut (*Cocos nucifera* L. (Arecaceae))-growing regions [1,2,3]. Keifer described the coconut mite in 1965 [4] based on samples gathered in the Mexican state of Guerrero. Coconut mite outbreaks typically occur during the dry season, which causes considerable financial losses for coconut producers [2,5]. Coconut mite colonies are concealed underneath the bracts of the fruits, where they feed on the meristematic tissue, which leads to surface scars, deformation, premature fruit drop, and reduced size, weight, albumen, water content, and yield [1,2,6,7]. Up to 60% of the crop can be lost when coconut mites are present [8], and a reduction in the copra yield or even the premature dropping of fruits can also occur [9,10]. Additionally, due to obvious necrosis brought on by this pest, the price of coconuts intended for the fresh market has been remarkably lowered [2,5].

Coconut is a palm plant grown in more than 90 countries, especially in tropical areas, with over 12 million hectares of land dedicated to its growth, and over 80% of its production is in Asia [11,12]. Thailand is the sixth largest coconut producer in the world [13], and approximately 80–90% of its coconut products are exported; as a result, the total cultivation area of coconuts in Thailand is approximately 19,840 hectares overall, and 320,000 tons of coconuts have been produced (Ministry of Commerce, Nonthaburi, Thailand) [12,14,15]. As Thailand is located in tropical and subtropical areas, its weather is favorable for coconut mites [2,13]. According to reports, the eriophyid *Colomerus novahebridensis* Keifer, which inhabits the same general habitat as *A. guerreronis*, i.e., the area between the bracts and the adjacent fruit surface, has occasionally caused considerable damage to Thailand’s coconut plantations [13,16]. Coconut mites have damaged coconuts in the Americas and Africa for over 40 years, and for the last 20 years, they have also arrived in Asian countries such as India and Sri Lanka; as a result, studies exploring coconut mite biology, ecology, taxonomy, management, and economic importance have been continually published [2,3,17,18]. Recently, coconut plantations in Thailand have had *A. guerreronis* outbreaks [19]. Moreover, *A. guerreronis* are known as vectors of plant pathogens [20], and they have been reported in the plant quarantine list of Thailand [19]. However, *A. guerreronis* has not been reported as a coconut pest in Thailand in previous research. Therefore, we aimed to identify *A. guerreronis* populations from five different major coconut production areas in Thailand via their morphology and molecular properties, and we also aimed to provide information that officials can use to prevent *A. guerreronis* infestations.

## 2. Materials and Methods

### 2.1. Coconut Mite Sampling and Identification

*A. guerreronis* specimens were collected in agricultural plots in 5 localities (Figure 1), located in the central, eastern, and western regions of Thailand.

Samples from 25 *A. guerreronis* populations were collected from coconut fruits produced in the area reported in Figure 1 and Table 1. Mites were collected by directly examining fruits under a stereomicroscope, preserving them in 99% absolute alcohol to perform molecular identification, and subsequently mounting them on slides in a modified Berlese medium [21]. Fifteen perfect-condition female specimens from each population were chosen for analysis and mounted in the dorsoventral position. The analyzed slide-mounted specimens were deposited in the mite reference collection of the Department of Entomology, Faculty of Agriculture at Kamphaeng Saen, Kasetsart University, Kamphaeng Saen Campus, Nakhon Pathom 73140, Thailand.

The mites were observed under a phase contrast light optical microscope (Leica DM100 LED) (100× objective). The morphology and nomenclature follow Lindquist [22] and the systematic classification is based on Amrine et al. [23]. The morphological characteristics essential for the determination of species were compared with the original description of this species [4]. The phase contrast optical microscope (Leica DM100 LED, Leica Microsystems Ltd., Heerbrugg, Switzerland) was linked to a digital camera (Leica MC170 HD), which was then linked to a computer to capture the images of the body regions of the chosen specimens. Images of the prodorsal shield and coxigenital region were obtained using a 100× magnification objective, and images of the ventral region were obtained using a 40× magnification objective. To conduct the landmark digitization, the prodorsal shield, coxigenital, and ventral sections of the *A. guerreronis* body were each individually assessed (including the coxigenital region and opisthosoma). These regions were selected because of their taxonomic importance and because a high number of landmarks could be defined. Ten landmarks in the prodorsal shield (Figure 2A), twelve in the coxigenital (Figure 2B), and nineteen in the ventral section (Figure 3) were selected. The classification of landmarks was based on [24,25]. Landmark data were produced with a series of programs called TpsUtil64 ver. 1.81 and Tpsdig264 ver. 2.32 software [26,27] and plotted. Deformation grids were obtained as thin-plate spline warps using MorphoJ software version 1.07a [28,29] and were plotted and used to explain deviations in the shape of each species from the average landmark configuration (consensus). Using a PCA of the covariance matrix of the population-averaged Procrustes coordinates, shape differences among the studied populations were further investigated. PCA was carried out using the MorphoJ program [28,30].

### 2.2. Molecular Identification

#### 2.2.1. Sample Collection

A total of 25 coconut mite samples were collected and stored at −20 °C. Thus, all subsequent extractions were performed with approximately 200 pooled adult mites. These samples were obtained from one breed distributed across 5 provinces in Thailand (Table 2). Animal handling and experimentation followed the animal experimental procedures and guidelines approved by the Ethics Committee of Kasetsart University (ID Code ACKU65-AGK-039).

#### 2.2.2. Total DNA Isolation

The total DNA was isolated from the coconut mites using the CTAB buffer method. As a first step, 0.6 mL of the CTAB buffer was added, and the buffer was homogenized with the sample. Then, the homogenized sample was incubated overnight at 65 °C, frozen for 15 min at 20 °C, and incubated for a further 15 min at 65 °C. Then, 0.5 mL of chloroform/isopropanol (24:1) was added, thoroughly mixed by shaking, and then incubated for 2 min at 25 °C. Samples were centrifuged at 12,000× *g* for 5 min. The DNA exclusively remained in the upper aqueous phase, and was transferred to a fresh tube. The DNA in the aqueous phase was precipitated by adding 0.5 mL of isopropyl alcohol. Then, the sample was incubated overnight at 4 °C. Afterwards, the supernatant was removed, the DNA pellet was air-dried, and then dissolved in 20 µL of RNase-free water before being stored at −20 °C. Finally, the DNA concentration and purity were assessed via gel electrophoresis using a NanoDrop Spectrophotometer 2000 (ThermoFisher Scientific, Waltham, MA, USA).

#### 2.2.3. PCR Amplification and Sequencing

The amount of DNA was multiplied using a PCR with a random hexamer primer. The PCR was amplified using the tree primer set of rDNA-ITS (F-rDNA-ITS: AGAGGAAGTAAAAGTCGTAACAAG and R-rDNA-ITS: ATATGCTTAAATTCAGGGGG [31]), 16S mtDNA (F-mtDNA-16S: CCGGTCTGAACTCAGATCACG and R-mtDNA-16S: CGCCTGTTTAACAAAAACAT) [32], and COI mtDNA (F-mtDNA-COI: GGATCACCTGATATAGCATTCCC and R-mtDNA-COI: CCCGGTAAAATT AAAATATAAACTTC [32]). PCR amplifications were performed using a Taq^®^ 2X Master Mix (New ENGLAND BioLabs^®^ inc., Ipswich, MA, USA) in a final volume of 25 µL containing 12.5 µL of 2X Master Mix, 1 µL of each primer (10 pmol/µL), 8.5 µL of dH2O, and 2 µL of DNA, which was multiplied with a random hexamer primer (100 ng/µL). The cycling profile included a 5 min preliminary denaturation cycle at 95 °C, followed by 40 denaturation cycles at 95 °C for 30 s, annealing at 50 °C for 30 s, and extension at 72 °C for 30 s, with a final extension at 72 °C for 5 min. The PCR products were separated via electrophoresis on a 2% agarose gel and visualized under ultraviolet light. Sequencing of the PCR product samples was carried out at ATGC Co., Ltd. (Ward Medic IDT, Bangkok, Thailand).

#### 2.2.4. Data Analysis

DNA sequences were manually checked using BioEdit [33] and then aligned using the ClustalW algorithm in MEGA 11.0 [34]. All the mtDNA16S sequences were aligned and trimmed to 400 and 600 bp, corresponding to the nucleotide positions (nps) 1–600 of the coconut mite (*A. guerreronis*) reference sequence DQ063558.1(USA), DQ063570.1(SrL), DQ063575.1(Ind), DQ063553.1(Br), DQ063564.1(Ben), DQ063562.1(Tanc), DQ063561.1(Ven), and DQ063560.1(Mexc). All the rDNA ITS sequences were aligned and trimmed to 987 bp, corresponding to the nucleotide positions (nps) 1–987 of the coconut mite (*A. guerreronis*) reference sequence DQ060624.1(Br), DQ060623.1(Ind), DQ060618.1(SrL), DQ060599.1(Tanc), DQ060597.1(Ben), DQ060582.1(Ven), DQ060580.1(USA), and DQ060576.1(Mexc). All the mtDNA COI sequences and trimmed to 494 bp, corresponding to the nucleotide positions (nps) 1–494 of the coconut mite (*A. guerreronis*) reference sequence MT253711.1, MT019905.1, and JX289538.1. The evolutionary history of the genomic DNA and mtDNA was inferred by using the maximum likelihood method, the Tamura 3-parameter model [35], and the Kimura 2-parameter model [36]. Initial trees for the heuristic search were obtained automatically by applying the Neighbor-Join and BioNJ algorithms to a matrix of pairwise distances, which were estimated using the Tamura 3-parameter model. The topology was then selected using the superior log likelihood value. Evolutionary analyses were conducted in MEGA11 [34].

## 3. Results

### 3.1. Coconut Mite Sampling and Identification

Based on comparing the morphological characteristics of the mite from our research with the original description of this species [4], we concluded that it was *Aceria guerreronis*. The symptoms of coconut fruits with *A. guerreronis* attack have yellowish-to-brownish triangular scars that start at the margin of the bracts and increase as the fruit grows (Figure 4).

### 3.2. Landmark-Based Morphometric Methods

Analyses were carried out in order to learn more about the morphological differences across *A. guerreronis* populations. The consensus shapes, based on the prodorsal shield of the 9 landmarks, the coxigenital area of the 12 landmarks, and the ventral regions of the 19 landmarks belonging to 75 *A. guerreronis* individuals from five localities, are shown in Figure 5A, Figure 6A and Figure 7A, respectively. Moreover, in Figure 5B–F, Figure 6B–F and Figure 7B–F, thin-plate spline deformation grids are depicted as variations in the *A. guerreronis* morphology from five locations in Thailand.

By applying a multivariate analysis (PCA), we discovered that the *A. guerreronis* populations from the sampled localities morphologically varied. The PCA performed on the shape coordinates of 19 landmarks in the ventral region of 75 specimens from the five studied populations resulted in 34 principal components, with the first two components explaining 51.75% of the total variation (PC1 29.29%, PC2 22.46%). Then, the PCA performed on the shape coordinates of nine landmarks in the prodorsal shields of 75 specimens from the five studied populations resulted in 16 principal components, with the first two components explaining 70.89% of the total variation (PC1 60.75%, PC2 10.14%). Moreover, the PCA performed on the shape coordinates of 12 landmarks in the coxigenital areas of 75 specimens from the five studied populations resulted in 20 principal components, with the first two components explaining 47.83% of the total variation (PC1 35.99%, PC2 11.84%). When we plotted the populations against their respective values for PRIN1 and PRIN2 (Figure 5, Figure 6, Figure 7 and Figure 8), we found that several mite populations, namely Ratchaburi (d) and Samut Sakhon (e), were dispersed along both axes, showing considerable morphometric diversity within each of them. On the other hand, mites from other populations, including Chachoengsao (a), Nakhon Pathom (b), and Pathum Thani (c), were concentrated in a very small area of the graphic, indicating a higher degree of morphometric similarity within each of those populations.

### 3.3. Molecular Identification

Based on the morphological characteristics, we initially identified the coconut mite strains (Table 2) as belonging to *A. guerreronis*. We recovered a total of 42 sequences for three genomic regions, including 10 for the rDNA ITS, 22 for the mtDNA 16S, and 10 for the mtDNA COI from the 25 *A. guerreronis* samples that we analyzed. Due to difficulties in obtaining the PCR amplification, which were likely brought on by the degraded state of some of the materials, we did not sequence every DNA template for the three DNA areas.

The tree with the highest log likelihood (−1445.42) is shown (Figure 9), and the percentage of trees in which the associated taxa clustered together is shown next to the branches. This analysis involved 18 nucleotide sequences. We included the 1st + 2nd + 3rd + Noncoding codon positions, which resulted in a total of 833 positions in the final dataset.

The percentage of trees in which the associated taxa clustered together is shown next to the branches. We used a discrete Gamma distribution to model the evolutionary rate differences among the sites (five categories (+G, parameter = 6.5149)). This analysis involved 30 nucleotide sequences. We included the 1st + 2nd + 3rd + Noncoding codon positions, which resulted in a total of 636 positions in the final dataset. The tree with the highest log likelihood (−4493.79) is shown (Figure 10).

The trees with the highest log likelihood (−2254.25) of the 600 bp PCR product and (−1278.80) of the 400 bp PCR product are shown (Figure 11). The percentage of trees in which the associated taxa clustered together is shown next to the branches. We automatically obtained an initial tree(s) for the heuristic search by applying the Neighbor-Join and BioNJ algorithms to a matrix of pairwise distances, which we estimated using the Tamura 3-parameter model and then selecting the topology with a superior log likelihood value. This analysis involved 19 nucleotide sequences. We included the 1st + 2nd + 3rd + Noncoding codon positions, and totals of 433 positions (600 bp PCR product) and 405 positions (400 bp PCR product) were present in the final dataset.

The tree with the highest log likelihood (−1353.95) is shown (Figure 12), and the percentage of trees in which the associated taxa clustered together is shown next to the branches. We used a discrete Gamma distribution to model evolutionary rate differences among the sites (five categories (+G, parameter = 1.7145)). The rate variation model allowed for some of the sites to be evolutionarily invariable ([+I], 8.39% sites). This analysis involved 13 nucleotide sequences. We included the 1st + 2nd + 3rd + Noncoding codon positions, which resulted in a total of 392 positions in the final dataset.

## 4. Discussion

We first reported and identified coconut mites in Thailand as *A. guerreronis* based on their morphological and molecular properties, which we obtained with multiple DNA sequences. We found that these methods were useful in supporting or augmenting the conventional morphology and that they enhanced the characterization and validation of genetic barcoding. Morphometric analyses can help researchers resolve the issue that conventional taxonomic approaches are insufficient to delimit morphologically identical eriophyoid mites that inhabit various plants. Because of this, many mites have been regarded as either different entities or as a single oligophagous species. In all existing studies, researchers have performed analyses similar to [2,37,38,39,40,41,42]. It should be emphasized that several morphological characteristics may help researchers distinguish between congeneric species or species that belong to various eriophyoid genera. In addition, morphological/morphometric and genetic variations in eriophyoid mites such as *Colomerus vitis* (Pagenstecher), the erineum strain (Eriophyidae) from grape vines, can differ according to geographical areas, sampling seasons, host plant physiology, and environmental factors [43,44].

After analyzing the morphological variability of the mites using geometric morphometric techniques on three body regions of *A. guerreronis* populations inhabiting five localities of Thailand, our general conclusions are as follows: The results of a principal component analysis (PCA) show generally comparable patterns for all population combinations from the various geographic regions. They clearly demonstrate that *A. guerreronis* had variations across its geographic distribution range. We anticipate that any of those factors could account for the morphological differences across *A. guerreronis* populations that resulted from the mites inhabiting coconut hosts in various geographical locations. In some cases, the results of the morphometric investigations show that environmental influences were not the primary variability determinants. Sometimes, according to the results of morphometric assessments, environmental factors were not the primary variability determinants. Following the findings of Navia et al. [25], some populations in neighboring Brazil from climatically comparable regions, as well as those from the northeastern coastal region, did exhibit very noticeable differences. This clearly implies that the observed physical variation in such groups is tightly correlated with the genetic background of the species.

The fact that the morphology of Eriophyoidea mites may be related to their habitat structure is well known [25,37,45]. We found that the *A. guerreronis* population collected from different localities in Thailand was morphologically similar to the geographically varied ventral regions’ closest populations (Chachoengsao, Nakhon Pathom, and Pathum Thani), but that it was different from Samut Sakhon and Ratchaburi.

The shape of Eriophyoidea mites may be related to their habitat structure. Hence, Navia et al. [25] investigated morphological variations in the ventral regions, prodorsal shield, and coxigenital area among *A. guerreronis* populations in America, Africa, and Asia. Their findings regarding the coxigenital and ventral areas confirmed the origin and invasion history of this species, which agreed with those obtained using molecular markers [46]. Furthermore, their finding that substantial morphological variations existed between American populations supported earlier claims that *A. guerreronis* originated in the United States, whereas the similar morphology between the African and Asian populations suggested a shared origin and rapid separation of those populations.

Researchers have recently reported that *Aceria guerreronis* is present in Asia, specifically in India and Sri Lanka [25,46,47]. The connection between the African and Asian populations reveals their related genetic characteristics, suggesting that the Asian populations resulted from an introduction from Africa. In Thailand, *A. guerreronis* is classified as a quarantine pest that is prohibited under the Plant Quarantine Act B.E. 2507 (1964) (No. 3) B.E. 2550 (2007), and there have never been reports of *A. guerreronis* in Thailand before. The results of our phylogenetic analyses on the 25 mite samples, which identify the mites that were closely related to mites from an earlier publication in India, suggest that both populations had a common origin.

The results of rDNA ITS and mtDNA COI sequences reveal that the *A. guerreronis* collected from Chachoengsao and Ratchaburi were closely related to those from an earlier publication in India [48]. Moreover, the results of the mtDNA 16s sequences not only show that the *A. guerreronis* collected from Samut Sakhon were closely related to those from earlier publications in Brazil, India, Sri Lanka, Tanzania, Benin, Venezuela, America, and Mexico, but they also show that Nakhon Pathom and Pathum Thani were closely related to those from an earlier publication on Samut Sakhon [48].

Recently, the Department of Agriculture (DOA) Thailand reported that the coconut mite, *A. guerreronis*, has been discovered in 18 provinces of Thailand, including Amnat Charoen, Chainat, Suphanburi, Kamphaeng Phet, Lop Buri, Nakhon Pathom, Nakhon Sawan, Nakhon Ratchasima, Pathum Thani, Phetchabun, Phichit, Phitsanulok, Ratchaburi, Saraburi, Sing Buri, Suphanburi, and Amnat Charoen; additionally, 4.2% of all the trees that were surveyed had fruit damage. The important coconut-growing regions of Thailand’s upper north, northeast, and south did not have this species. The results of this study reveal that *A. guerreronis* is also present in two provinces, Chachoengsao and Samut Sakhon, in addition to Nakhon Pathom, Pathum Thani, and Ratchaburi.

Other Asian and Pacific locations where coconut mites have not yet been detected may be at risk for experiencing coconut mite infestations. This is because the primary way coconut mites are introduced to remote locations entails the movement of any propagation tissue from palm trees, as well as the transit or exchange of propagation host plant material, especially from Thailand, which represents a quarantine risk.

In conclusion, the coconut mite *A. guerreronis* is not only a serious threat to coconut plantations, but it is also a quarantine pest in Thailand. We isolated coconut mites from infested plants and identified them based on their morphological characteristics and molecular properties, which we obtained using multiple DNA sequences. We are the first to detail a geometric morphometric analysis and molecular identification of *A. guerreronis* in Thailand. To discover efficient biological control measures, researchers must determine the historical distribution of the mite. Knowing the population spread patterns is also economically important because the coconut mite still poses a threat to Asian nations where it has not yet been found. Understanding its spread could help scholars predict the likelihood of future incursions and could help them direct quarantine measures to stop the spread of the pest.

## Figures and Tables

**Figure 1 insects-13-01022-f001:**
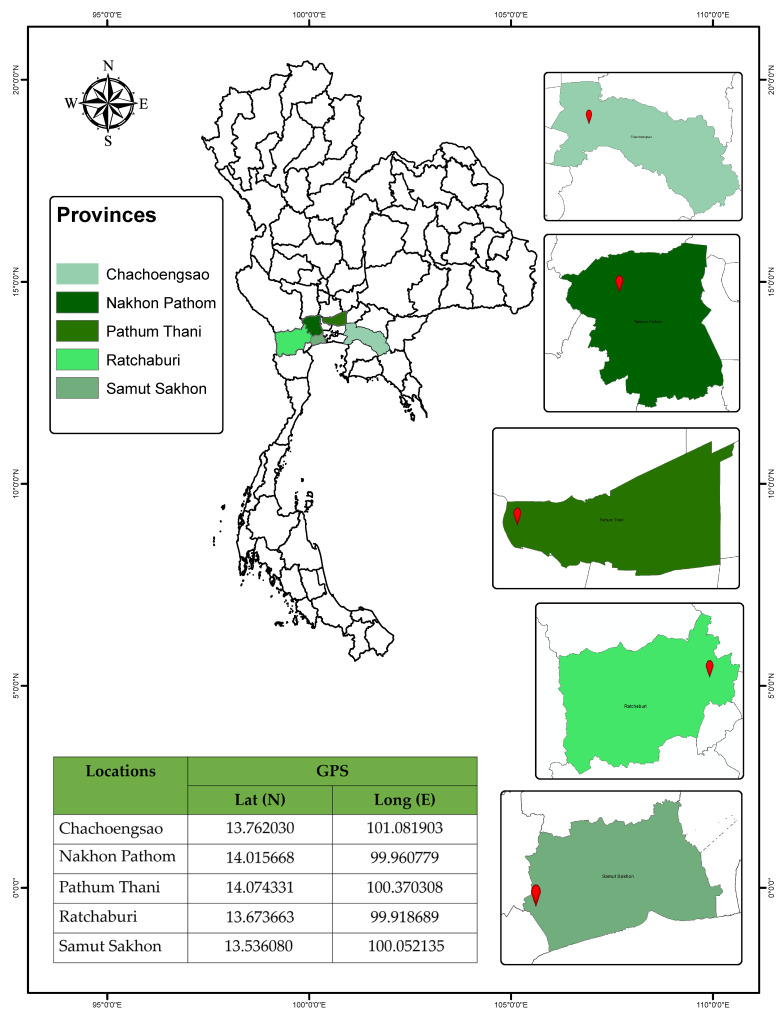
Studied area of coconut mite *A. guerreronis* distribution in Thailand.

**Figure 2 insects-13-01022-f002:**
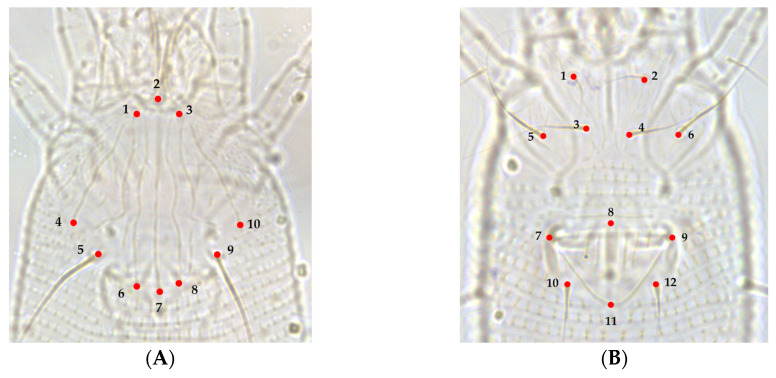
Morphological landmarks selected from prodorsal shield and coxigenital region of *A. guerreronis* according to [24,25]: (**A**) prodorsal shield, (1) left base of frontal lobe, (2) anterior central tip of front lobe, (3) right base of frontal lobe, (4) extremity of left lateral line on shield posterior margin, (5) base of left scapular seta (*sc*), (6) extremity of left admedian line on shield posterior margin, (7) extremity of median line on shield posterior margin, (8) extremity of right admedian line on shield posterior margin, (9) base of right scapular seta (*sc*), (10) extremity of right lateral line on shield posterior margin; (**B**) coxigenital region, (1) base of left coxal seta I (*lb*), (2) base of right coxal seta I (*lb*), (3) base of left coxal seta II (*la*), (4) base of right coxal seta II (*la*), (5) base of left coxal seta III (*2a*), (6) base of right coxal seta III (*2a*), (7) left joint of the anterior portion of genital seta (*3a*) tubercle with the anterolateral margin of coverflap, (8) projection of anterocentral tip of coverflap, (9) right joint of the anterior portion of genital seta (*3a*) tubercle with the anterolateral margin of coverflap, (10) base of left genital seta (*3a*), (11) posterocentral tip of epigynium, (12) base of right genital seta (*3a*).

**Figure 3 insects-13-01022-f003:**
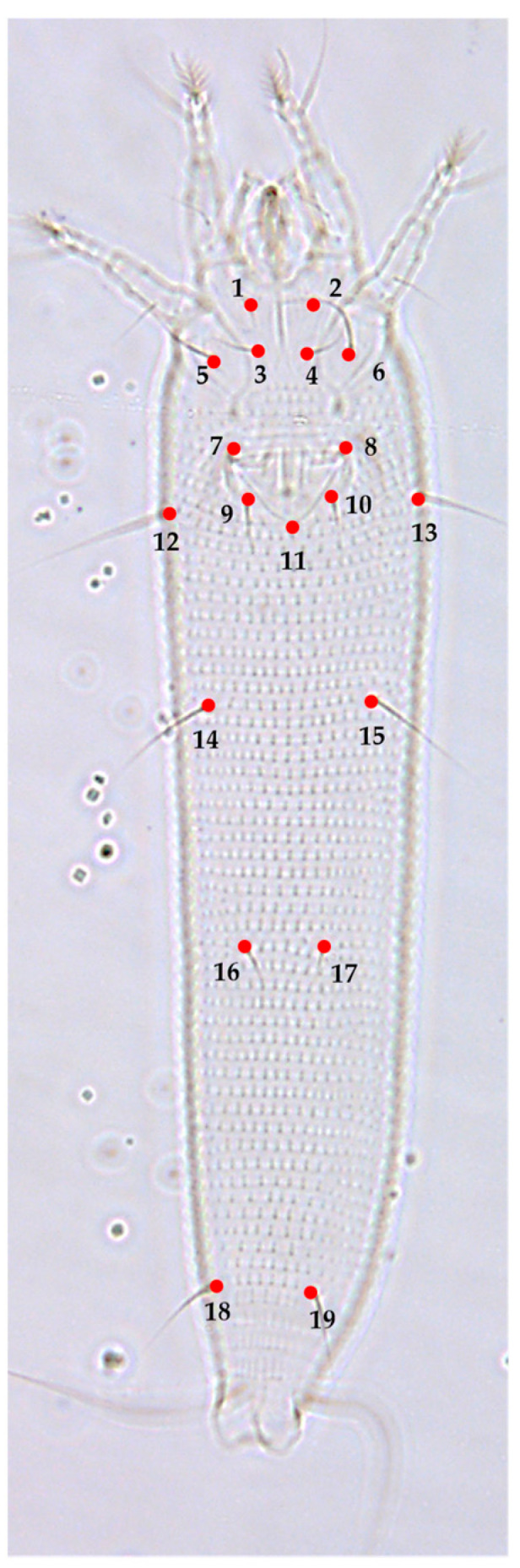
Morphological landmark selected from the ventral region according to [24,25]: ventral region, (1) base of left coxal seta I (*lb*), (2) base of right coxal seta I (*lb*), (3) base of left coxal seta II (*la*), (4) base of right coxal seta II (*la*), (5) base of left coxal seta III (*2a*), (6) base of right coxal seta III (*2a*), (7) left joint of the anterior portion of genital seta *(3a)* tubercle with the anterolateral margin of coverflap, (8) right joint of the anterior portion of genital seta (*3a*) tubercle with the anterolateral margin of coverflap, (9) base of left genital seta (*3a*), (10) base of right genital seta (*3a*), (11) posterocentral tip of coverflap, (12) base of left lateral seta (*c2*), (13) base of right lateral seta (*c2*), (14) base of left ventral seta I (*d*), (15) base of right ventral seta I *(d*), (16) base of left ventral seta II (*e*), (17) base of right ventral seta II (*e*), (18) base of left ventral seta III (*f*), (19) base of right ventral seta III (*f*).

**Figure 4 insects-13-01022-f004:**
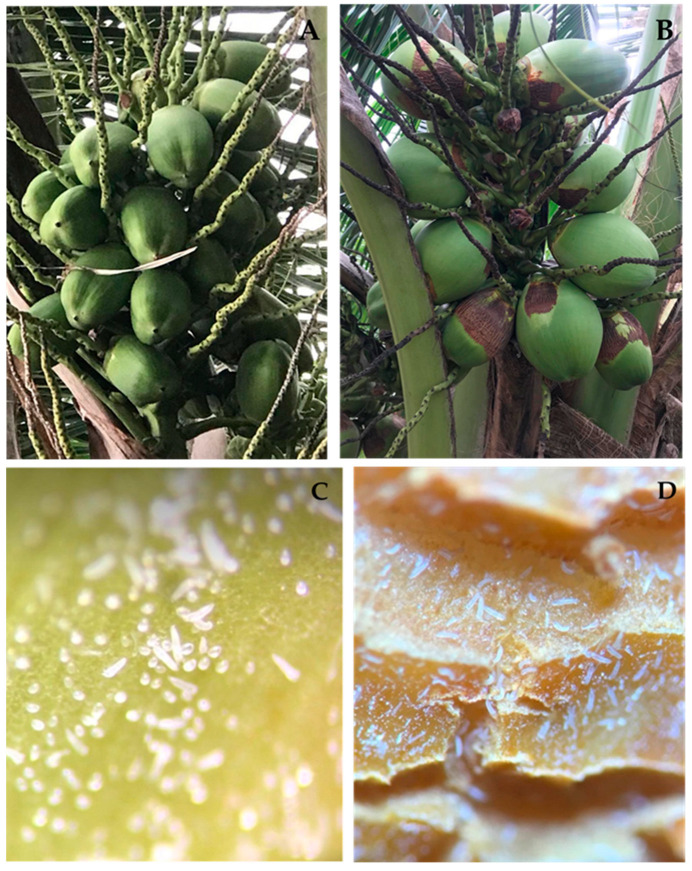
Most of the nuts older than 8 weeks showed dried-up attack symptoms caused by *A. guerreronis*: (**A**) eight-week-old nuts without infestation; (**B**) nut showing spread-out attack symptoms and premature fruit drops; (**C**) different life stages of *A. guerreronis* from the mite-infested nut below tepals and (**D**) on nut.

**Figure 5 insects-13-01022-f005:**
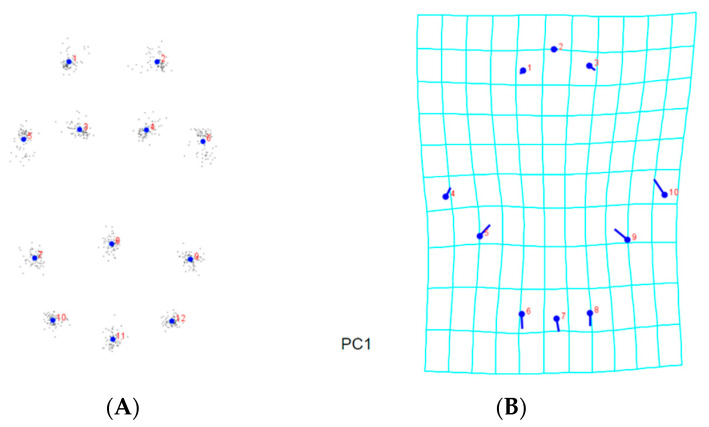
Thin-plate spline visualization of coxigenital area: (**A**) consensus, (**B**) Chachoengsao (C), (**C**) Nakhon Pathom (N), (**D**) Pathum Thani (P), (**E**) Ratchaburi (R), (**F**) Samut Sakhon (S).

**Figure 6 insects-13-01022-f006:**
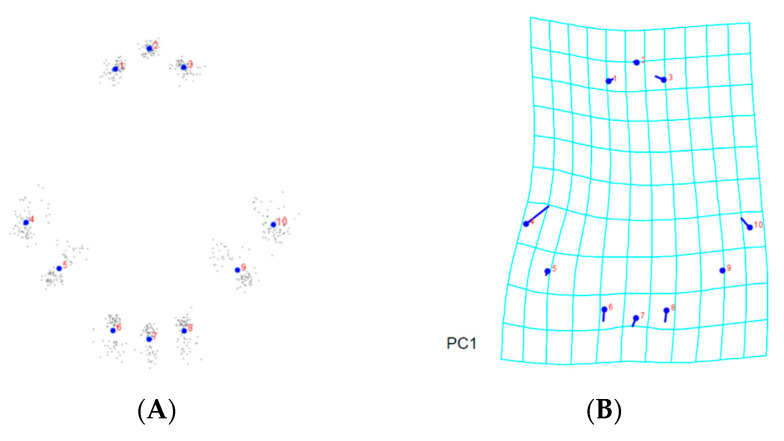
Thin-plate spline visualization of prodorsal shield: (**A**) landmark consensus, based on superimposition of 75 coxigenital regions, (**B**) Chachoengsao (C), (**C**) Nakhon Pathom (N), (**D**) Pathum Thani (P), (**E**) Ratchaburi (R), (**F**) Samut Sakhon (S).

**Figure 7 insects-13-01022-f007:**
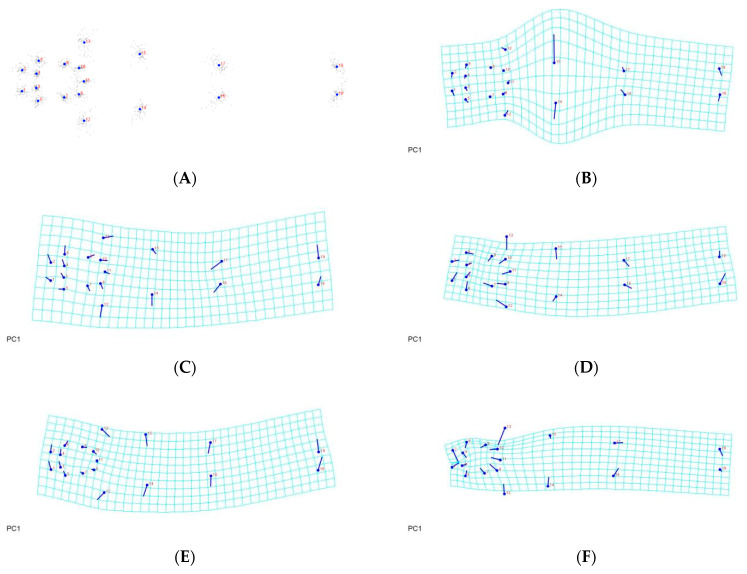
Thin-plate spline visualization of ventral side: (**A**) landmark consensus, based on superimposition of 75 ventral region, (**B**) Chachoengsao (C), (**C**) Nakhon Pathom (N), (**D**) Pathum Thani (P), (**E**) Ratchaburi (R), (**F**) Samut Sakhon (S).

**Figure 8 insects-13-01022-f008:**
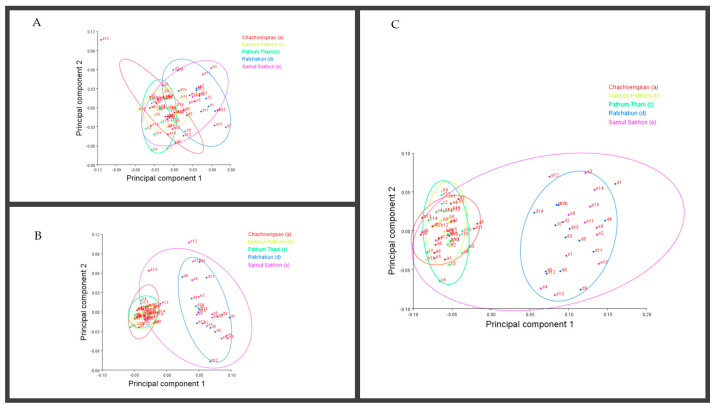
Principal component analysis of Thailand populations of *A. guerreronis*. Individuals plotted against their values for the first two principal components. Letters represent collection sites specified in Table 1: (**A**) ventral region, (**B**) coxigenital, and (**C**) prodorsal shield.

**Figure 9 insects-13-01022-f009:**
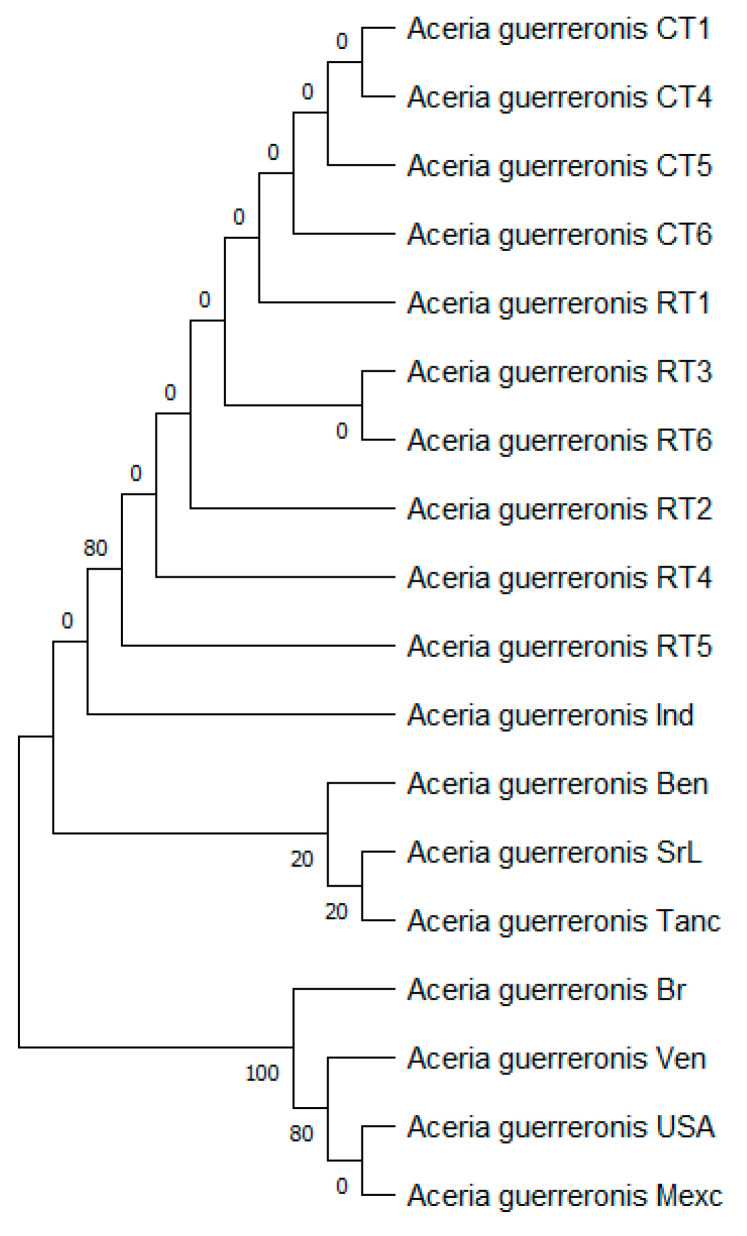
Evolutionary analysis by maximum likelihood method of rDNA ITS gene.

**Figure 10 insects-13-01022-f010:**
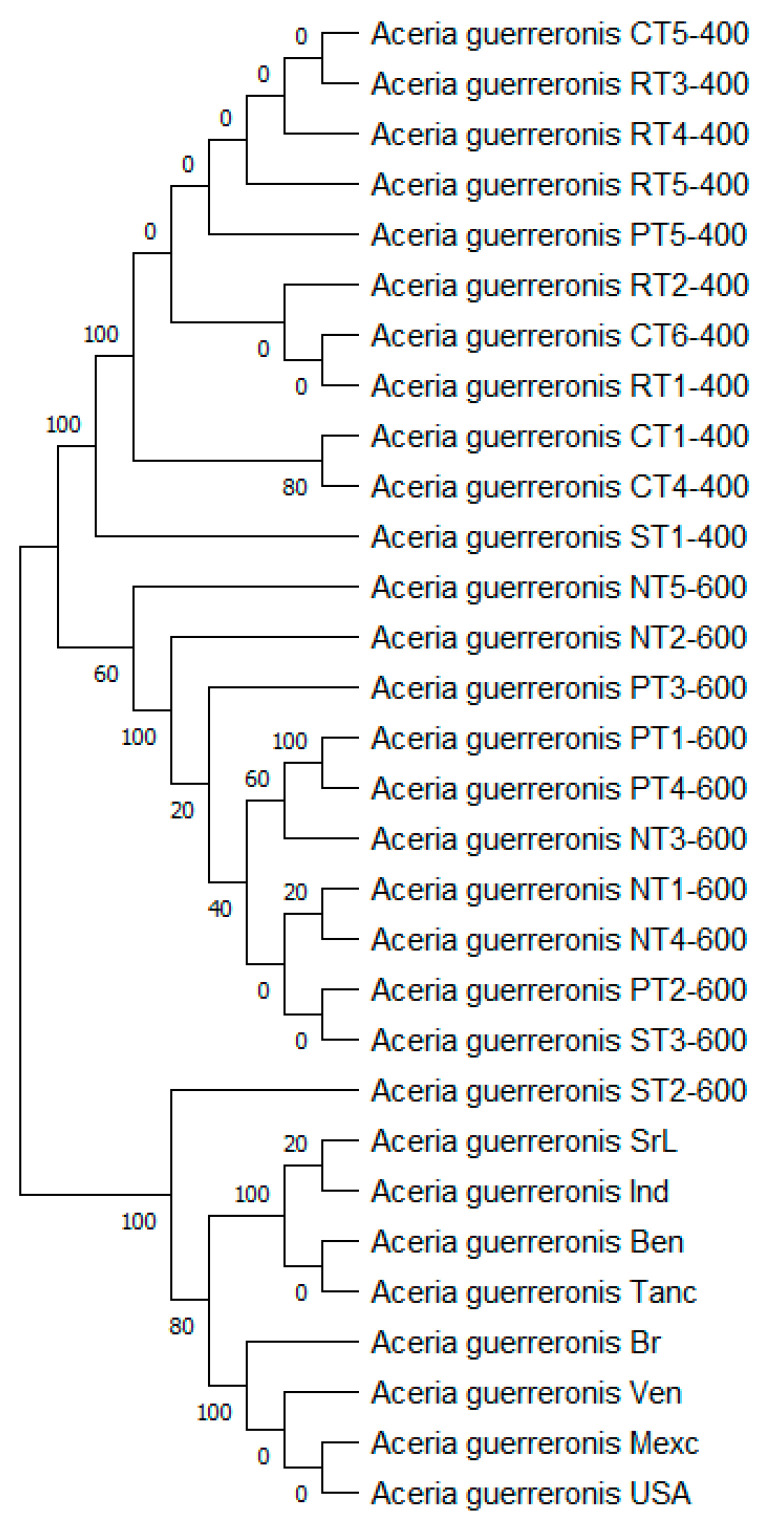
Evolutionary analysis by maximum likelihood method of mtDNA 16s gene.

**Figure 11 insects-13-01022-f011:**
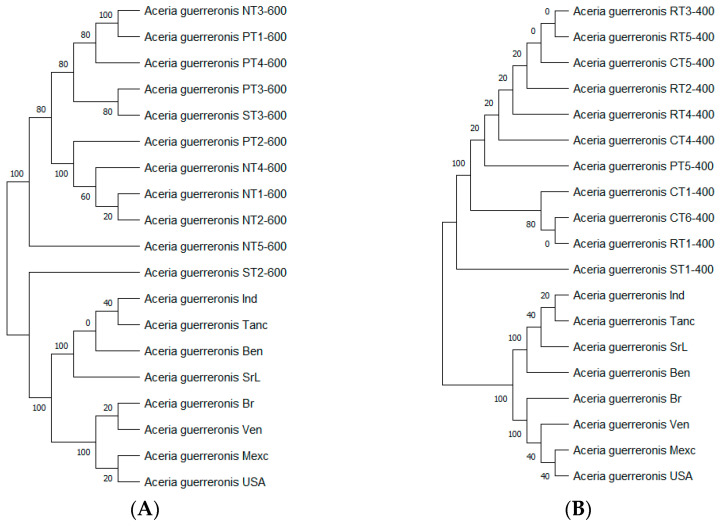
Evolutionary analysis by maximum likelihood method of mtDNA 16s gene (600 bp PCR product (**A**), 400 bp PCR product (**B**)).

**Figure 12 insects-13-01022-f012:**
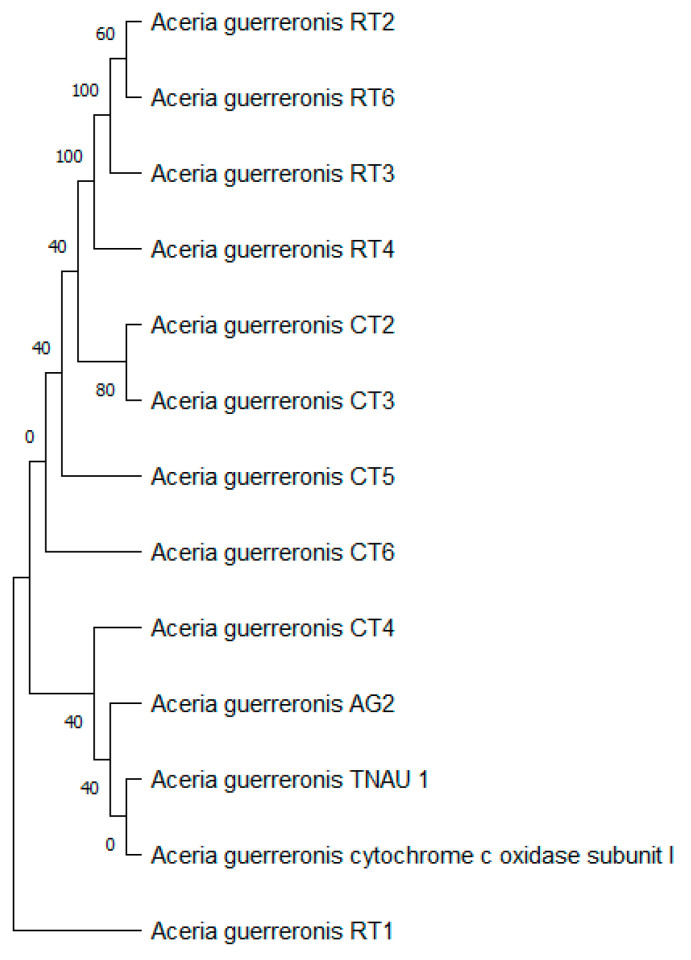
Evolutionary analysis by maximum likelihood method of mtDNA COI.

**Table 1 insects-13-01022-t001:** Sampling localities of *A. guerreronis* populations. Codes and numbers of measured females.

Continent	Country	Locality	Code	Number of Females
Asia	Thailand	Chachoengsao	a	15
		Nakhon Pathom	b	15
		Pathum Thani	c	15
		Ratchaburi	d	15
		Samut Sakhon	e	15

**Table 2 insects-13-01022-t002:** Coconut mites collected from provinces in Thailand.

Samples	Total Coconut Mite Samples
Chachoengsao (C)	CT1, CT2, CT3, CT4, CT5, CT6 (6 samples)
Nakhon Pathom (N)	NT1, NT2, NT3, NT4, NT5 (5 samples)
Pathum Thani (P)	PT1, PT2, PT3, PT4, PT5 (5 samples)
Ratchaburi (R)	RT1, RT2, RT3, RT4, RT5, RT6 (6 samples)
Samut Sakhon (S)	ST1, ST2, ST3 (3 samples)

## Data Availability

The DNA sequence data obtained in this study were deposited in GenBank, and the accession numbers are as follows: rDNA ITS: CT1 (OP325554), CT4 (OP325555), CT5 (OP325556), CT6 (OP325557), RT1 (OP325558), RT2 (OP325559), RT3 (OP325560), RT4 (OP325561), RT6 (OP325562); mtDNA 16s (400 bp PCR product): CT1 (OP361287), CT4 (OP361288), CT5 (OP361289), CT6 (OP361290), PT5 (OP361291), RT1 (OP361292), RT2 (OP361293), RT3 (OP361294), RT4 (OP361295), RT5 (OP361296), ST1 (OP361297); 16s (600 bp PCR product): NT1 (OP326138), NT2 (OP326139), NT3 (OP326140), NT4 (OP326141), NT5 (OP361286), PT1 (OP326142), PT2 (OP326143), PT3 (OP326144), PT4 (OP326145), ST2 (OP326146), S3 (OP326147); and COI: CT2 (OP351383), CT3 (OP351384), CT4 (OP351385), CT5 (OP351386), CT6 (OP351387), RT2 (OP351388), RT6 (OP351389).

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
