# Peer review of "Geometric Morphometric Analysis and Molecular Identification of Coconut Mite, Aceria guerreronis Keifer (Acari: Eriophyidae) Collected from Thailand"

_insects, 2022, doi:10.3390/insects13111022_

Round 1

Reviewer 1 Report

Dear Authors,

I have now completed my review of "Geometric morphometric analysis and molecular identification of coconut mite, Aceria guerreronis Keifer (Acari: Eriophyidae) collected from Thailand." for Insects, and submitted my conclusion, 'Reconsider after major revision'.

My remarks and suggestions for correction and completion can be seen in the manuscript. 

Author Response

I wish to thanks reviewer #1 for his/her appreciation of our work and also for pertinent comments which certainly will improve readability of the manuscript. We agree with the reviewer with all sections following the comments and already had corrected it with yellow-highlight.

after I corrected following your kind comments then I submitted it to English editing from MDPI please recheck from Insects-1963277-R-Clean-English-edited-52402 version

Sincerely Yours,

Suradet Buttachon

Reply to Reviewers ‘comments

            - I have now completed my review of "Geometric morphometric analysis and molecular identification of coconut mite, Aceria guerreronis Keifer (Acari: Eriophyidae) collected from Thailand." for Insects, and submitted my conclusion, 'Reconsider after major revision'.

My remarks and suggestions for correction and completion can be seen in the manuscript.

Reply: I wish to thanks reviewer #1 for his/her appreciation of our work and also for pertinent comments which certainly will improve a readability of the manuscript. We agree with the reviewer with all section following the comments and already had corrected it with yellow-highlight.

- English language and style

(x) Extensive editing of English language and style required

Reply:  We agree with the reviewer and already had edited it by English editing from MDPI.

Sincerely Yours,

Suradet Buttachon

Reviewer 2 Report

The results of the work are significant and interesting, BUT Material and methods and Results and must be presented in a better way.

When these two chapters are done better, only then can the discussion chapter be reviewed

The all part of Material and methods about morphometric is necessary write again.  Written like this is unclear and beside that there is no the part of method of landmark and what statistics were used.

Please see the next manuscripts:

Denise Navia• Cecı´lia B. S. Ferreira• Aleuny C. Reis, Manoel G. C. Gondim Jr. Traditional and geometric morphometrics supporting the differentiation of two new Retracrus (Phytoptidae) species associated with heliconias. Exp Appl Acarol (2015) 67:87–121.DOI 10.1007/s10493-015-9934-z

Navia D, Moraes GJ, Querino RB (2006) Geographic variation in the coconut mite, Aceria guerreronis Keifer (Acari: Eriophyidae): a geometric morphometric analysis. Int J Acarol 32:301–314.

Biljana Vidovic,  Vida Jojic´  Ivana Maric´ Slavica Marinkovic,  Richard Hansen •Radmila Petanovic. Geometric morphometric study of geographic and hostrelated variability in Aceria spp. (Acari: Eriophyoidea) inhabiting Cirsium spp. (Asteraceae). Exp Appl Acarol (2014) 64:321–335. DOI 10.1007/s10493-014-9829-4

The results of morphometric analysis are unclearly and they are not systematized. It is necessary to write everything again. Please see the next manuscripts:

Denise Navia• Cecı´lia B. S. Ferreira• Aleuny C. Reis, Manoel G. C. Gondim Jr. Traditional and geometric morphometrics supporting the differentiation of two new Retracrus (Phytoptidae) species associated with heliconias. Exp Appl Acarol (2015) 67:87–121.DOI 10.1007/s10493-015-9934-z

Biljana Vidovic,  Vida Jojic´  Ivana Maric´ Slavica Marinkovic,  Richard Hansen •Radmila Petanovic. Geometric morphometric study of geographic and hostrelated variability in Aceria spp. (Acari: Eriophyoidea) inhabiting Cirsium spp. (Asteraceae). Exp Appl Acarol (2014) 64:321–335. DOI 10.1007/s10493-014-9829-4

In the part 3.2 there are duplicated the passage from material and methods. In the results of molecular identification write only results, that is, explain the given Figures 9,10, 11, 12.

See also the comments in the text

Author Response

I wish to thanks reviewer #2 for his/her appreciation of our work and also for pertinent comments which certainly will improve readability of the manuscript. We agree with the reviewer with all sections following the comments and already had corrected it with green-highlight.

after I corrected following your kind comments then I submitted it to English editing from MDPI please recheck from Insects-1963277-R-Clean-English-edited-52402 version

Sincerely Yours,

Suradet Buttachon

Reviewer 3 Report

The paper needs to be carefully revised for the language. There are some mistyping, grammar errors and often sentences confuse for the reader like me. This makes the paper hard to be understood in some points. You need to make substantial improvements in the presentation and accuracy of the language. I would suggest that you ask a native English speaker to read through your manuscript. For example, in the Simple summary you write that you investigated the geometric morphometric analysis and molecular identification and I guess that an analysis cannot be investigated and an identification, as well. There are many other sentences which are vague or written in a way that are difficult to understand.

One further general comment regards the use of references. Sometimes they are missed and sometimes it seems that you mentioned papers just collecting data/information from previous papers. The original paper should be mentioned. In addition, sometimes are reported more references, but I guess that just one could be mentioned.

In the Abstract, 25 populations are indicated for the morphological and genetic study, but there is not correspondence with what is reported in Materials and methods.

The hypothesis that there are morphological/morphometric variations of the populations collected in different sites is appreciated, but these variations can have also other causes. In fact, the morphometry of the same population can be influenced by the sampling time (see VALENZANO D., TUMMINELLO M.T., GUALANDRI V., DE LILLO E., 2020 – Morphological and molecular characterization of the Colomerus vitis erineum strain (Trombidiformes: Eriophyidae) from grapevine erinea and buds. - Exp. Appl. Acarol., 80: 183-201.), which means that the environmental factors and the plant physiology can be involved, as well as the host genotype (see Javadi Khederi et al. 2014). These aspects cannot be avoided into the Discussion paragraph.

Pay attention on the paragraph Conclusions. It seems to be completely unuseful and some conclusions are in the Discussion paragraph. It is suggested to delete Conclusions paragraph or join it with Discussion paragraph.

SIMPLE SUMMARY

L24-25 How the results of this study can functional studiers? That sentence is too vague and unclear.

L25-26 How the results of this study can be helpful for the pest management strategies. Also this sentence is too vague.

KEY WORDS

Acari is in the title.

ABSTRACT

L36 Aceria becomes A.; Keifer has to be deleted; Prostigmata has to be substituted by Acari.

1. INTRODUCTION

L48 replace Insect with mite

2. MATERIALS AND METHODS

L104 One paper by Petanovic (PETANOVIĆ R.U., 2016 – Towards an integrative approach to taxonomy of Eriophyoidea (Acari, Prostigmata) an overview. - Ecol. Mont., 7: 580-599) clearly discusses the traits which are more representative for the alpha-taxonomical studies of the eriophyoids. I suggest to give a look and mention that paper.

L117, 120 what do you mean? Do you mean outer submedian lines?

L124, 126, 127 replace epigynium with coverflap.

Caption of Figure 3 seems to be wrong.

2.4.2. Total DNA extraction. It is unclear how many sample were processed and if they were pooled together or singly treated.

L163-4 It is an extended reference and not a number corresponding to a reference.

Title of 2.4.4 seems to be wrong.

3. RESULTS 3.1 Symptom recognition

Is this title matching the content? I do not believe so.

L247 what does Table 2 mean?

L253-293 show many repetitions of what reported in 2. Materials and methods.

L318 which method do you mean?

L329 you mention that “the following general conclusions” but you are in the paragraph entitled Discussion.

L366-372 references should be added

L389 infected is not used for mites.

Author Response

I wish to thanks reviewer #1 for his/her appreciation of our work and also for pertinent comments which certainly will improve readability of the manuscript. We agree with the reviewer with all sections following the comments and already had corrected it with blue-highlight.

after I corrected following your kind comments then I submitted it to English editing from MDPI please recheck from Insects-1963277-R-Clean-English-edited-52402 version

Sincerely Yours,

Suradet Buttachon

Reply to Reviewers ‘comments

Reviewer 3 following comments and suggestions

One further general comment regards the use of references. Sometimes they are missed and sometimes it seems that you mentioned papers just collecting data/information from previous papers. The original paper should be mentioned. In addition, sometimes are reported more references, but I guess that just one could be mentioned.

Reply: I wish to thanks reviewer 3 for his/her comments and for detecting various error. These comments certainly help improve the readability of this manuscript. We agree with the reviewer and already had corrected it.

The paper needs to be carefully revised for the language. There are some mistyping, grammar errors and often sentences confuse for the reader like me. This makes the paper hard to be understood in some points. You need to make substantial improvements in the presentation and accuracy of the language. I would suggest that you ask a native English speaker to read through your manuscript. For example, in the Simple summary you write that you investigated the geometric morphometric analysis and molecular identification and I guess that an analysis cannot be investigated and an identification, as well. There are many other sentences which are vague or written in a way that are difficult to understand.

- English language and style

(x) Extensive editing of English language and style required

 Reply:  We agree with the reviewer and already had edited it by English editing from MDPI.

In the Abstract, 25 populations are indicated for the morphological and genetic study, but there is not correspondence with what is reported in Materials and methods.

Reply: 25 populations from 5 provinces in Thailand are indicated for the morphological and genetic study following Table 1 and 2 as mentioned in Materials and methods. We agree with the reviewer and already had added it with blue-highlight in the attached file.

The hypothesis that there are morphological/morphometric variations of the populations collected in different sites is appreciated, but these variations can have also other causes. In fact, the morphometry of the same population can be influenced by the sampling time (see VALENZANO D., TUMMINELLO M.T., GUALANDRI V., DE LILLO E., 2020 – Morphological and molecular characterization of the Colomerus vitis erineum strain (Trombidiformes: Eriophyidae) from grapevine erinea and buds. - Exp. Appl. Acarol., 80: 183-201.), which means that the environmental factors and the plant physiology can be involved, as well as the host genotype (see Javadi Khederi et al. 2014). These aspects cannot be avoided into the Discussion paragraph.

Reply: We agree with the reviewer and already had added it discussion paragraph with blue-highlight in the attached file.

Pay attention on the paragraph Conclusions. It seems to be completely unuseful and some conclusions are in the Discussion paragraph. It is suggested to delete Conclusions paragraph or join it with Discussion paragraph.

Reply:  We agree with the reviewer but we would like to keep conclusion paragraph as following the journal template. We also had edited it in order to useful and understandable our work and give a suggestion. We need to keep main idea of our work and point out important of our work so that it might be similarity mean with discussion paragraph.

SIMPLE SUMMARY

L24-25 How the results of this study can functional studiers? That sentence is too vague and unclear.

Reply: Our results of this study can functional studiers as this is the first report geometric morphometric analysis and molecular identification of A. guerreronis in Thailand. A. guerreronis is quarantine pest in Thailand so our result can be data base of future functional study as another countries where it has not yet been reported about A. guerreronis.

L25-26 How the results of this study can be helpful for the pest management strategies. Also this sentence is too vague.

 Reply: Our results of this study can be helpful for the pest management strategies following,

  1. As we mention guerreronis is quarantine pest in Thailand and we found out it infested coconut plantation in Thailand. For pest management strategies firstly, we need to know first what kind of pest and where its origin and history then we can find out the solution to control it. For example, we would like to use biological control which are natural enemies as predators for controlling A. guerreronis. There are a lot of research about these but still not successful even the same pest as A. guerreronis so it is expected that effective natural enemies of a pest will be found in the place of origin can be use in that area and mostly stable solution as I think.
  2. Aa we mention in findings would be helpful in designing pest management strategies via “quarantine pests in Thailand “ . The mostly important, we need to find out the solution to eliminate and prove quarantine measures to stop the spread of the guerreronis in Thailand if not our coconut fruit from Thialand cannot export and will have economic problem. The coconut fruit is an important commercial fruit crop ranking fourth of Thai fruit crops with highest export values. Our results of this study can be helpful to know exactly this is a coconut mite, A. guerreronis by geometric morphometric analysis and molecular identification because coconut plantation in Thailand have problem with other eriophyiod mite as Colomerus novahebridensis and make us confused about these coconut mite can damage coconut fruit.

KEY WORDS

Acari is in the title.

Reply: We agree with the reviewer and already had corrected it with blue-highlight

ABSTRACT

L36 Aceria becomes A.; Keifer has to be deleted; Prostigmata has to be substituted by Acari.

Reply: We agree with the reviewer and already had corrected it

  1. INTRODUCTION

L48 replace Insect with mite

Reply: We agree with the reviewer and already had corrected it

  1. MATERIALS AND METHODS

L104 One paper by Petanovic (PETANOVIĆ R.U., 2016 – Towards an integrative approach to taxonomy of Eriophyoidea (Acari, Prostigmata) an overview. - Ecol. Mont., 7: 580-599) clearly discusses the traits which are more representative for the alpha-taxonomical studies of the eriophyoids. I suggest to give a look and mention that paper.

Reply: We agree with the reviewer and already had added it following your regard with blue-highlight in the attached file.

L117, 120 what do you mean? Do you mean outer submedian lines?

Reply: Yes, I mean that outer submediant lines and the morphological landmarks description followed the procedures of References 21 and 22. and both are study of Aceria guerreronis which similar with our main study.

  1. Bookstein, F.L. Morphometrics in evolutionary biology: the geometry of size and shape change, with examples from fishes; Academy of Natural Sciences of Philadelphia: 1985; Volume 15.
  2. Navia, D.; Moraes, G.d.; Querino, R. Geographic pattern of morphological variation of the coconut mite, Aceria guerreronis Keifer (Acari: Eriophyidae), using multivariate morphometry. Brazilian Journal of Biology 2009, 69, 773-783, doi:https://doi.org/10.1590/S1519-69842009000400004

L124, 126, 127 replace epigynium with coverflap.

Reply: We agree with the reviewer and already had corrected it

Caption of Figure 3 seems to be wrong.

Reply: We agree with the reviewer and already had corrected it following your regard with blue-highlight in the attached file.

2.4.2. Total DNA extraction. It is unclear how many sample were processed and if they were pooled together or singly treated.

Reply: As we mention in 2.4. Molecular identification, 2.4.1. Sample collection

A total of 25 coconut mite samples were collected and stored at -20°C……………………..”

-Each sample was singly treated following Table 2.

L163-4 It is an extended reference and not a number corresponding to a reference.

Reply: We agree with the reviewer and already had corrected it

Title of 2.4.4 seems to be wrong.

Reply: We agree with the reviewer and already had corrected it

  1. RESULTS 3.1 Symptom recognition

Is this title matching the content? I do not believe so.

Reply: We agree with the reviewer and already had corrected it as 3.1 Coconut Mite Sampling and Identification

L247 what does Table 2 mean?

Reply: We agree with the reviewer and already had corrected it

L253-293 show many repetitions of what reported in 2. Materials and methods.

Reply: We agree with the reviewer and already had corrected it following your regard with blue-highlight in the attached file.

L318 which method do you mean?

Reply: We agree with the reviewer and already had corrected it with blue-highlight

L329 you mention that “the following general conclusions” but you are in the paragraph entitled Discussion.

Reply:  We agree with the reviewer and already had corrected it.

L366-372 references should be added

Reply:  We agree with the reviewer and already had added it.

L389 infected is not used for mites.

Reply:  we already had corrected it.

Sincerely Yours,

Suradet Buttachon

Round 2

Reviewer 1 Report

Dear Authors,

I have now completed my review of "Geometric morphometric analysis and molecular identification of coconut mite, Aceria guerreronis Keifer (Acari: Eriophyidae) collected from Thailand." for Insects, and submitted my conclusion, 'Accept after minor revision'.

Several references are not cited in the text. My corrections and completions can be seen in the proof.

Author Response

Dear Reviewer 1

I wish to thanks reviewer 1 for his/her comments and for detecting various error. These comments certainly help improve the readability of this manuscript.

I hereby wish to submit the manuscript entitled “Geometric Morphometric Analysis and Molecular Identification of Coconut Mite, Aceria guerreronis Keifer (Acari: Eriophyidae) Collected from Thailand” which was recommended for minor revisions required.

In revising the manuscript, I have taken into consideration the recommendations and suggestions provided by 3 reviewers.

The manuscript I have uploaded is now named Insects-1963277-R. To facilitate 3 reviewers and the Editor, I have uploaded the version whose corrections have been using Track changes version (Insects-1963277-R) following your suggestions to the system.

Sincerely Yours,

Suradet Buttachon

Lecture of Entomology

Department of Entomology, Faculty of Agriculture at Kamphaeng Saen,

Kasetsart University, Kamphaeng Saen Campus,

Nakhon Pathom, 73140

E-mail: fagrsdbu@ku.ac.th

Reply to Reviewers ‘comments

            - I have now completed my review of "Geometric morphometric analysis and molecular identification of coconut mite, Aceria guerreronis Keifer (Acari: Eriophyidae) collected from Thailand." for Insects, and submitted my conclusion, 'Accept after minor revision'.

Several references are not cited in the text. My corrections and completions can be seen in the proof.

Reply: I wish to thanks reviewer #1 for his/her appreciation of our work and also for pertinent comments which certainly will improve a readability of the manuscript. We agree with the reviewer with all section following the comments and already had corrected it.

Sincerely Yours,

Suradet Buttachon

Reviewer 2 Report

Now the manuscript is better. Some small additions should be made to make everything precise and clear.

All comment can see in the text.

Author Response

Dear Reviewer 2

I wish to thanks reviewer 2 for his/her comments and for detecting various error. These comments certainly help improve the readability of this manuscript.

I hereby wish to submit the manuscript entitled “Geometric Morphometric Analysis and Molecular Identification of Coconut Mite, Aceria guerreronis Keifer (Acari: Eriophyidae) Collected from Thailand” which was recommended for minor revisions required.

In revising the manuscript, I have taken into consideration the recommendations and suggestions provided by 3 reviewers.

The manuscript I have uploaded is now named Insects-1963277-R. To facilitate 3 reviewers and the Editor, I have uploaded the version whose corrections have been using Track changes version (Insects-1963277-R) following your suggestions to the system.

Sincerely Yours,

Suradet Buttachon

Lecture of Entomology

Department of Entomology, Faculty of Agriculture at Kamphaeng Saen,

Kasetsart University, Kamphaeng Saen Campus,

Nakhon Pathom, 73140

E-mail: fagrsdbu@ku.ac.th

Reviewer 3 Report

I have to express my sincere congratulations to the authors for the successful efforts they did for improving the papers following almost all suggestions reported in the first run of my revision.

The paper has still a few aspects which need to be improved. I would like to stress again the care in using references.

Concerning the DNA extraction, from figs 9-12 I understand that mite samples were treated singly. But, how many mites composed each sample? It is suggested to give information on this. In addition, you refer that ITS region was sequenced. Which one?

Again the paragraph Conclusions seems to be completely unuseful and some conclusions are in the Discussion paragraph. I come back suggesting to delete Conclusions paragraph or join it with the Discussion paragraph.

L22 add (Acari: Eriophyidae) after Keifer

L34 which ITS region was studied (1 or 2)?

L48 the reference code number should be added after 1965 and should regard the Keifer’s paper.

L53 are references 5 and 6 appropriate here?

L55 is reference 8 appropriate here?

L57 is references 4 appropriate here?

L61 are references 10, 11 and 12 appropriate here?

L64 are references 11, 13 and 14 appropriate here?

L66 is references 12 appropriate here?

L66 add Keifer after novahebridensis

L74 is vector of one or more plant pathogens? A reference should be added here.

Fig. 1 should be enlarged. It is too small

LL91-93 It should become: samples from 25 A. guerreronis populations were collected from coconut fruits produced in the area reported in Fig. 1 and Table 1. Mites were collected through directly examining fruits under a stereomicroscope, preserving them …..

L96 add “slide mounted” between analysed and specimens

L137 replace epigynium with coverflap

L145 replace epigynium with coverflap

L268 I counted 22 sequences for mtDNA 16S

L271 add DNA between three and areas.

LL272-275 It should be specified that you are talking of ITS region.

LL281-300 It should be specified that you are talking of mtDNA.

LL305-306 It should be specified that you are talking of COI

L321 add “to” between researchers and resolve

L328 add (Pagenstecher) after vitis

L329 delete Trombidiformes:

L334 you mention “Conclusions” whereas you are in the Discussion paragraph. On the same line there is a “:”.

LL365-368 Pay attention to the singular and plural cases of words and verbs.

LL368-370 The sentence is unclear

References: pay attention to the italic style of the species and genus names

Author Response

Dear Reviewer 3

I wish to thanks reviewer 3 for his/her comments and for detecting various error. These comments certainly help improve the readability of this manuscript.

I hereby wish to submit the manuscript entitled “Geometric Morphometric Analysis and Molecular Identification of Coconut Mite, Aceria guerreronis Keifer (Acari: Eriophyidae) Collected from Thailand” which was recommended for minor revisions required.

In revising the manuscript, I have taken into consideration the recommendations and suggestions provided by 3 reviewers.

The manuscript I have uploaded is now named Insects-1963277-R. To facilitate 3 reviewers and the Editor, I have uploaded the version whose corrections have been using Track changes version (Insects-1963277-R) following your suggestions to the system.

Sincerely Yours,

Suradet Buttachon

Lecture of Entomology

Department of Entomology, Faculty of Agriculture at Kamphaeng Saen,

Kasetsart University, Kamphaeng Saen Campus,

Nakhon Pathom, 73140

E-mail: fagrsdbu@ku.ac.th

Reply to Reviewers ‘comments

Reviewer 3 following comments and suggestions

I have to express my sincere congratulations to the authors for the successful efforts they did for improving the papers following almost all suggestions reported in the first run of my revision.

Reply: I wish to thanks reviewer 3 for his/her comments and for detecting various error. These comments certainly help improve the readability of this manuscript.

The paper has still a few aspects which need to be improved. I would like to stress again the care in using references.

Reply: We agree with the reviewer and already had corrected it

Concerning the DNA extraction, from figs 9-12 I understand that mite samples were treated singly. But, how many mites composed each sample? It is suggested to give information on this. In addition, you refer that ITS region was sequenced. Which one?

Reply: We agree with the reviewer and already had corrected it as all subsequent extractions were done with approximately 200 pooled adult mites. In addition, I refer ITS region was sequenced from this Reference 31………….we also mention it  in Line 173-174 primer set of rDNA-ITS (F-rDNA-ITS: AGAGGAAGTAAAAGTCGTAACAAG and R-rDNA-ITS: ATATGCTTAAATTCAGGGGG [31]),

Ben Ali, Z.; Boursot, P.; Said, K.; Lagnel, J.; Chatti, N.; Navajas, M. Comparison of Ribosomal ITS Regions Among Androctonus spp. Scorpions (Scorpionida: Buthidae) from Tunisia. Journal of Medical Entomology 2000, 37, 787-790, doi:10.1603/0022-2585-37.6.787.

Again the paragraph Conclusions seems to be completely unuseful and some conclusions are in the Discussion paragraph. I come back suggesting to delete Conclusions paragraph or join it with the Discussion paragraph.

Reply: We disagree with the reviewer, and we would like to keep Conclusions paragraph to following the template of MDPI journal. It need to write separate with Discussion paragraph. Moreover.

L22 add (Acari: Eriophyidae) after Keifer

Reply: We agree with the reviewer and already had added it

L34 which ITS region was studied (1 or 2)?

Reply: , ITS region was sequenced from this Reference 31………….we also mention it  in Line 173-174 primer set of rDNA-ITS (F-rDNA-ITS: AGAGGAAGTAAAAGTCGTAACAAG and R-rDNA-ITS: ATATGCTTAAATTCAGGGGG [31]),

Ben Ali, Z.; Boursot, P.; Said, K.; Lagnel, J.; Chatti, N.; Navajas, M. Comparison of Ribosomal ITS Regions Among Androctonus spp. Scorpions (Scorpionida: Buthidae) from Tunisia. Journal of Medical Entomology 2000, 37, 787-790, doi:10.1603/0022-2585-37.6.787.

L48 the reference code number should be added after 1965 and should regard the Keifer’s paper.

Reply: We agree with the reviewer and already had added it

L53 are references 5 and 6 appropriate here?

Reply: Yes, they are. We need to cite these references

L55 is reference 8 appropriate here?

Reply: Yes, it is. We need to cite this reference

L57 is references 4 appropriate here?

Reply: Yes, it is. We need to cite this reference

L61 are references 10, 11 and 12 appropriate here?

Reply: Yes, they are. We need to cite these references

L64 are references 11, 13 and 14 appropriate here?

Reply: Yes, they are. We need to cite these references

L66 is references 12 appropriate here?

Reply: Yes, it is. We need to cite this reference

L66 add Keifer after novahebridensis

Reply: We agree with the reviewer and already had added it

L74 is vector of one or more plant pathogens? A reference should be added here.

Reply: We agree with the reviewer and already had added it

Fig. 1 should be enlarged. It is too small

Reply: We agree with the reviewer and already had corrected it

LL91-93 It should become: samples from 25 A. guerreronis populations were collected from coconut fruits produced in the area reported in Fig. 1 and Table 1. Mites were collected through directly examining fruits under a stereomicroscope, preserving them …..

Reply: We agree with the reviewer and already had corrected it

L96 add “slide mounted” between analysed and specimens

Reply: We agree with the reviewer and already had added it

L137 replace epigynium with coverflap

Reply: We agree with the reviewer and already had corrected it

L145 replace epigynium with coverflap

Reply: We agree with the reviewer and already had corrected it

L268 I counted 22 sequences for mtDNA 16S

Reply: We agree with the reviewer and already had corrected it

L271 add DNA between three and areas.

Reply: We agree with the reviewer and already had added it

LL272-275 It should be specified that you are talking of ITS region.

Reply: We agree with the reviewer and already had added it

LL281-300 It should be specified that you are talking of mtDNA.

Reply: We agree with the reviewer and we already specified ……is shown (Figure 10).

The percentage of trees in which the associated taxa clustered together is shown next to the branches. We used a discrete Gamma distribution to model the evolutionary rate differences among the sites (five categories (+G, parameter = 6.5149)). This analysis involved 30 nucleotide sequences. We included the 1st+2nd+3rd+Noncoding codon positions, which resulted in a total of 636 positions in the final dataset. The tree with the highest log likelihood (-4493.79) is shown (Figure 10).

LL305-306 It should be specified that you are talking of COI

Reply: We agree with the reviewer and we already specified ……is shown (Figure 12).

The tree with the highest log likelihood (-1353.95) is shown (Figure 12)…………………………

L321 add “to” between researchers and resolve

Reply: We agree with the reviewer and already had added it

L328 add (Pagenstecher) after vitis

Reply: We agree with the reviewer and already had added it

L329 delete Trombidiformes:

Reply: We agree with the reviewer and already had deleted it

L334 you mention “Conclusions” whereas you are in the Discussion paragraph. On the same line there is a “:”.

Reply: We apologize to said that I do not understand what the reviewer comment……….As we mention ………………………our general conclusions are as follows:…………………….in order to Discussion.

LL365-368 Pay attention to the singular and plural cases of words and verbs.

Reply: We agree with the reviewer and already had corrected it.

LL368-370 The sentence is unclear

Reply: We agree with the reviewer and already had edited English by MDPI

“In Thailand, A. guerreronis is classified as quarantine pests that are prohibited under the Plant Quarantine Act B.E. 2507 (1964) (No. 3) B.E. 2550 (2007), and there has never been reports of A. guerreronis in Thailand before. The results of our phylogenetic analyses on the 25 mite samples, which identified the mites that were closely related to mites from an earlier publication in India, suggested that both populations had a common origin”

References: pay attention to the italic style of the species and genus names

Reply: We agree with the reviewer and already had corrected it.

Sincerely Yours,

Suradet Buttachon